# Comparative Analysis of How the Fecal Microbiota of Green-Winged Saltator (*Saltator similis*) Diverge among Animals Living in Captivity and in Wild Habitats

**DOI:** 10.3390/ani14060937

**Published:** 2024-03-19

**Authors:** Larissa Caló Zitelli, Gabriela Merker Breyer, Mariana Costa Torres, Luiza de Campos Menetrier, Ana Paula Muterle Varela, Fabiana Quoos Mayer, Cláudio Estêvão Farias Cruz, Franciele Maboni Siqueira

**Affiliations:** 1Laboratório de Bacteriologia Veterinária, Universidade Federal do Rio Grande do Sul, Porto Alegre 91540-000, Brazil; larissazitelli@gmail.com (L.C.Z.); gabibreyer@hotmail.com (G.M.B.); mariana.exs@gmail.com (M.C.T.); luizamenetrier@gmail.com (L.d.C.M.); 2Programa de Pós-Graduação em Ciências Veterinárias, Universidade Federal do Rio Grande do Sul, Porto Alegre 91540-000, Brazil; 3Centro de Biotecnologia, Universidade Federal do Rio Grande do Sul, Porto Alegre 91501-970, Brazil; anapaulamut@gmail.com (A.P.M.V.); bimmayer@gmail.com (F.Q.M.); 4Centro de Estudos em Manejo de Aves Silvestres—CEMAS, Universidade Federal do Rio Grande do Sul, Porto Alegre 91540-000, Brazil; claudio.cruz@ufrgs.br

**Keywords:** bacteriome, captive animals, wild animals, microbiota, bacteria biomarker, songbirds, NGS

## Abstract

**Simple Summary:**

*Saltator similis* is a species of songbird that is a victim of illegal trafficking, native to Brazilian forests, and kept in captivity. Nine fecal samples were collected from free-living birds, and nine birds in captivity were sampled. Total bacterial DNA was obtained from the feces and sequenced. The most predominant phyla were analyzed and compared. The bacterial genera “Candidatus *Arthromitus*”, *Acinetobacter*, *Kocuria*, and *Paracoccus* were identified exclusively in animals living in captivity, which may be potential biomarkers associated with birds in captive environments and under a restricted diet and stressful lifestyle. This study presents the first description of the fecal bacterial community composition of *S. similis* living in two different lifestyles. Finally, our results suggest that the lifestyle of *S. similis* birds significantly impacts the composition of their fecal microbiota. The results can bring about new discussions about the management and health of captive birds.

**Abstract:**

The microbiota’s alteration is an adaptive mechanism observed in wild animals facing high selection pressure, especially in captive environments. The objective of this study is to compare and predict the potential impact of habitat on the fecal bacterial community of *Saltator similis*, a songbird species that is a victim of illegal trafficking, living in two distinct habitats: wild and captivity. Nine wild and nine captive *S. similis* were sampled, and total bacterial DNA was obtained from the feces. Each DNA sample was employed to the amplification of the V4 region of the 16S rDNA following *high*-throughput sequencing. The most predominant phyla in all songbirds, irrespective of habitat, were *Firmicutes*, *Bacteroidota*, *Proteobacteria*, and *Actinobacteriota.* Interestingly, a microbiota profile (phylogenetic and abundance relationship) related to habitat was identified. The genera “Candidatus *Arthromitus*”, *Acinetobacter*, *Kocuria*, and *Paracoccus* were exclusively identified in animals living in captivity, which can be a potential biomarker associated with birds in captive environments. This study presents the first description of the fecal bacterial community composition of *S. similis* living two different lifestyles. Finally, our results suggest that the lifestyle of *S. similis* birds significantly impacts the composition of the fecal microbiota. The animals living in captivity showed dysbiosis in the microbiota, with some bacteria genera being indicated as biological markers of environmental behavior. Thus, the present research provides a new concept of life quality measure for songbirds.

## 1. Introduction

The adaptability of an individual’s microbiota is a rapid mechanism for adjusting to new environments or diets, increasing the host’s chances of successful adaptation [1]. The adaptation of the intestinal microbiota is a tool observed in wild animals that quickly need changes to confront the adversities of environments with high selection pressure [2].

Songbirds are more sensitive to changes in their environments and diets due to their low weight and fast metabolism [3]. The urban environment brings major changes in their behavior and strong impacts on their health, such as the inaction of toxic gases [4] and loud noises [5], in addition to having to face common challenges such as parasites [6].

Previous studies that have analyzed the microbiota of birds, comparing wild birds with urbanized birds, detected in White-crowned Sparrow (*Zonotrichia leucophrys*) a significant decrease in their intestinal microbiota. The authors suggested that the loud noises present in urban centers affect not only the behavior of animals but their microbiota [5]. A work developed with Darwin’s Finches parasitized by vampire flies, in the wild or urbanized, concluded that urbanized animals have a strong impact on their microbiota, while urbanized and parasitized animals showed very low bacterial diversity and variety when compared to others [6].

An environment with high selection pressure for wild animals is the captive environment, as it forces them to change their diet and behavior, leading to constant stress [7]. Stress induces metabolic and physiological changes that also cause significant alterations in the host’s bacterial community [8]. These changes in the intestinal microbiota significantly impact the individual’s life and may even affect their nutritional condition, allowing the emergence of infectious conditions [7,8].

The Green-winged saltator (*Saltator similis*) is a popular songbird species in Brazil [9] known for its territorial behavior and harmonious song. These traits make it a sought-after species for bird contests, contributing to the trade of the species and unfortunately making it a victim of illegal trafficking [10,11]. Consequently, the population of *S. similis* living in captivity is notably high [11]. Hence, the objective of this study was to describe and compare the bacterial community comprising the fecal microbiota of *S. similis* in two distinct habitats: the wild and captivity. Additionally, this study aimed to comprehend the potential impact of habitat on the fecal microbiota of these animals.

## 2. Materials and Methods

### 2.1. Animal Selection, Capture, and Collection of Fecal Samples

The manipulation of animals was previously approved by the Animal Ethics Committee of the Federal University of Rio Grande do Sul (No. 23644) and licensed by the Chico Mendes Institute for Biodiversity Conservation (ICMBio) under license number 37567.

Nine wild and nine captive *S. similis*, adults and apparently healthy, were sampled in the spring of the year of 2022. The captive songbirds had been held by the Wild Animal Triage Center of the Regional Superintendence of the Brazilian environmental agency IBAMA (Instituto Brasileiro do Meio Ambiente e dos Recursos Naturais Renováveis), located in the municipality of Porto Alegre, Rio Grande do Sul state (30°17′28” S/51°18′04” W), Southern Brazil (Figure 1). This center is a legal unit responsible for receiving, identifying, evaluating, recovering, rehabilitating, and placing wild animals [11]. Overnight fecal samples from the captive birds were obtained by covering their cage tray with a plastic film. The fecal samples were stored in a new and sterile plastic tube and were kept refrigerated during transport to the laboratory. The wild *S. similis* were captured at the final daylight hour, as described earlier [12], kept overnight in a holding bag, and released in the capture site at dawn. Fecal samples from wild *S. similis* were collected from Barra do Ribeiro (30°20′59” S/51°14′44” W) and Eldorado do Sul (30°04′37” S/51°35′49” W) (approximate coordinates) municipalities in Rio Grande do Sul state, Southern Brazil (Figure 1).

### 2.2. Extraction of Nucleic Acids of Fecal Samples from Saltator Similis

The fecal samples were homogenized by vortex, and 5g was subjected to metagenomic DNA extraction following the protocol of the extraction kit Dneasy Powersoil (Qiagen, Hilden, Germany). Additionally, blank DNA extraction was performed as negative control (the kit reagents were subjected to DNA isolation) for further decontamination of the libraries.

The obtained metagenomic DNA was analyzed by spectrometry with Nanodrop (Thermo Fisher Scientific, Waltham, MA, USA) and fluorometry with Qubit 2.0 fluorometer (Thermo Fischer Scientific, Massachusetts, USA) to measure the metagenomic DNA concentration and quality, respectively. DNA samples were stored at −80 °C until use.

### 2.3. Amplification and Sequencing of the Bacterial Community of Fecal Samples

The V4 region of the 16S rDNA gene was amplified using the universal primers 515F 5′-GTGCCAGCMGCCGCGGTAA-3′ and 806R 5′-GGACTACHVGGGTWTCTAAT-3′ [13] with the Illumina adapter sequences attached to the 5′ end.

Reactions were prepared in 50 µL of mix containing 10× buffer, 0.2 mM dNTP, 2.0 mM MgSO_4_, 0.5 µM of primer forward and reverse, and 1 U of Platinum Taq DNA Polymerase High Fidelity (Invitrogen, Massachusetts, USA), using as template 100 ng of each metagenomic DNA from fecal samples. The cycles condition used was an initial denaturation at 94 °C for 3 min, followed by 25 cycles of 94 °C for 30 s, 55 °C for 30 s and 72 °C for 30 s, and a final extension at 72 °C for 3 min.

The amplicons were purified, the libraries were constructed with the Illumina MiSeq v2 500-cycle kit (250 bp paired-end reads), and the sequencing was performed on the Illumina MiSeq platform (Illumina, San Diego, CA, USA). For each animal, one library was generated and sequenced.

### 2.4. Taxonomic Analysis of the Bacterial Community from Fecal Samples

The analysis of the bacterial communities was performed using Quantitative Insights into Microbial Ecology 2 (QIIME2) version 2019.7 [14]; initially, the raw reads’ quality was assessed by FastQC software (v0.11.9), followed by low-quality sequences (Phred < 30), short length reads (<50 nt), and primer and adaptor sequences trimming, using the plugin q2-dada2- with pipeline DADA2 Callahan [15]. Amplicon sequence variants (ASVs) were annotated using both the Scikitlearn system and the SILVA 136 database [16]. The amplicon ASVs obtained from the DADA2 pipeline were merged into a single feature table using the q2-feature-table plugin. The ASVs were aligned with MAFFT (q2 alignment) [17] and used to build a phylogeny with fasttree2 (q2-filogenia) [18].

To remove contamination reads from the 16S-rDNA samples’ libraries, we used microDecon package [19] in RStudio v. 2021.09.0 (RStudio Team, 2015) based on the library of the blank control. Eukaryote, archaea, chloroplast, mitochondria, and unknown sequences were removed from further analyses. After filtering and decontaminations, the remaining percentage, of at least 50% non-chimeric reads, was with a minimum size of 2000 and a maximum of 300,000 bp.

### 2.5. Statistical Analyses of the Taxonomic Results

The bacterial communities were compared considering the habitat, i.e., songbirds from the wild vs. captivity. The data were imported from the QIIME2 environment to RStudio. The statistical analyses were performed using the package Microbiome v1.6.0 [20] and the package Phyloseq v1.28.0 RStudio [21]. To perform the alpha diversity analysis, the non-parametric Wilcoxon test was used [22] via the Vegan R package [23]. The beta diversity analysis was performed with a permutational multivariate analysis of variance [24] using the distance of the matrix obtained by principal coordinate analysis (PCoA) with permutational variance analysis test (PERMANOVA), implemented as the Adonis role in the Vegan R package [23].

The evolutionary distribution by abundance of genus was performed by the packages TREEIO and Phyloseq in RStudio, followed by abundance plotting of reads on a phylogenetic tree by the TREEIO package and Phyloseq package in RStudio.

The effect of lifestyle on the fecal microbiota of the analyzed animals was determined by microbiomeMarker package in RStudio with comparisons of the fecal microbiota from captive vs. wild *S. similis*.

## 3. Results

### 3.1. Diversity Metrics of the Fecal Samples’ Bacterial Communities

In the present study, we analyzed nine S. similis living in natural reserves (wild) and nine S. similis living in captivity, for a total of 18 fecal samples from adult and apparently health songbirds. Initially, the generated libraries underwent decontamination using blank control reads. Following quality control steps, the 18 libraries produced a total of 3,477,217 raw reads, of which 2,766,195 (79.55%) reads remained after quality filtering (Table 1).

The rarefaction curves of the 18 sequenced libraries showed that the ASVs, although satisfactorily represented in all samples, exhibited differences in richness among them (Appendix A). An intrinsic characteristic of the samples is illustrated in Appendix A, where the libraries were plotted based on sequence sample size and species richness. Additionally, libraries were plotted according to the reads’ sample sizes and density samples in Appendix A. These data suggest that bacterial diversity was effectively explored and is representative of the community present in each analyzed sample.

Statistical analyses were performed to identify the global differences in the composition of the fecal bacterial community between the two lifestyles (habitats) of *S. similis*, comparing the fecal bacterial communities of songbirds from captive and wild environments (Figure 2 and Figure 3). According to the Shapiro–Wilk normality test, there were reads differences in relation to richness and evenness (W = 0.71655; *p* = 0.0001252) in animals from each analyzed habitat. The rarefaction curves illustrated a higher richness in the bacterial community from wild songbirds’ feces (Appendix A). However, based on alpha diversity, no significant differences were observed (*p* > 0.5) in richness and evenness of the bacterial communities in both groups (captivity and wild animals) (Figure 2). Furthermore, beta diversity measured by the Qualitative Unweighted UniFrac (Figure 3a) and Quantitative Weighted UniFrac (Figure 3b) indicated no significant differences among bacterial populations neither in captivity nor in wild habitats. On the other hand, Bray–Curtis analysis (Figure 3c) indicated a distinct clustering between samples from captive and wild animals, suggesting specific fecal bacterial populations for each *S. similis* lifestyle.

### 3.2. Taxonomic Profile, Relative Abundance, and Differential Abundance of Bacterial Community Present in the Feces from Captive and Wild Saltator Similis

Taxonomic profile at the phyla, families, and genera levels detected in the samples was thoroughly explored to ensure a robust identification and comparison of the habitat effect (captivity and wild) on the fecal bacterial populations of *S. similis*. Differential abundance analysis, considering the habitats of *S. similis*, demonstrated no distinct phyla composition pattern in the analyzed animals (Figure 4a). The most predominant phyla in all songbirds, irrespective of habitat, were Firmicutes, Bacteroidota, Proteobacteria, and Actinobacteriota (Figure 4a). Upon visually analyzing the abundance of phyla based on habitat, a tendency toward a higher presence of Firmicutes in the wild group compared to the captivity group can be observed (Figure 4a).

At the family and genus levels, there was a significant variation in bacterial abundances of each population (Figure 4b,c). However, when qualitatively compared, the captivity group showed more diversity in the abundance of families and genera identified in the animals compared to the wild group (Figure 4b,c). In general, the most predominant bacterial families identified in the fecal samples were equally distributed between the habitats (Appendix A). Currently, the most abundant families between the habitats were Clostridiaceae, Campylobacteriaceae, and Catelliococcaceae (Figure 4b), while the genera that stood out regardless of habitat were “Candidatus *Arthromintus*” and *Campylobacter* (Figure 4c).

Appendix A illustrates the phyla, families, and genera that were identified with statistical significance when the taxa of both captive and wild groups were compared. The results highlighted a substantial identification of taxa common to both analyzed habitats. However, some important bacterial families and genera were observed as exclusive to animals either in captivity or the wild (Appendix A). Captive songbirds had the genera *Aeromonas*, *Acinetobacter, Empedobacter*, *Flavobacterium,* “Candidatus *Arthromintus*”, *Sphingobacterium*, and *Acidibacter* as exclusive to this habitat. In contrast, the genera *Catelliococcus*, *Actinobacillus*, *Brevibacterium*, *Clostridium sensu stricto* 1, *Serratia*, and *Mycoplasma* were observed only in wild songbirds. Finally, the genera *Anaerosporobacter* and *Campylobacter* were equally abundant in animals from both habitats.

### 3.3. Fecal Bacterial Community Profile Is Guided by the Habitat

The evolutionary distribution based on genus abundance resulted in a clustering of the genera according to their phylogenetic relationships (Figure 5). Furthermore, the analysis indicated that fecal genera exhibit notable evolutionary and abundance profiles based on the habitat of the animals. Some genera exclusively abundant in wild animals, including *Actinomyces*, *Helcobacillus*, *Brevibacterium*, *Mycoplasma*, and *Ureaplasma*, formed an isolated clade in the phylogenetic tree without genera shared with captive animals (Figure 5).

We also examined the enriched taxa in songbirds in captivity and from the wild as potential genus markers for the habitat to which the animals were subjected (Table 2). In captive songbirds, four fecal bacterial genera were identified as potential markers (*p* < 0.05): “Candidatus *Arthromitus*”, Acinetobacter, Kocuria, and Paracoccus. On the other hand, among the studied wild *S. similis*, no bacterial genus was detected as a marker.

## 4. Discussion

This study presents, for the first time, the fecal bacterial community of songbirds, specifically the *S. similis* species, in two distinct lifestyles: wild and captive. To date, no similar studies with this animal species have been published. The fecal microbiomes of songbirds from both habitats were very similar at phyla and family levels. However, at the genus level, important differences were identified, which could serve as indicators of animal health, given the presence of bacteria previously associated with dysbiosis or observed in sick animals, particularly in captive songbirds (Figure 4). The prevalence of certain genera in animals from the captive habitat indicates that the fecal bacterial community exhibits specific characteristics depending on the host’s origin. Additionally, the results highlight that captive animals have genus markers that could be used as indicators of stress conditions.

The exclusive identification of “Candidatus *Arthromitus*” in animals living in captivity is noteworthy. Although it has been described in the microbiome of passerines such as *Luscinia megarhynchos* and *Luscinia luscinia* [25], the relationship of this genus with hosts remains uncertain. While beneficial interactions of “Candidatus *Arthromitus*” have been noted, such as its potential use as a probiotic in poultry for meat production [26], its abundance is potentially higher in hosts experiencing intense or prolonged stress conditions or during infectious diseases [27]. Evidence from studies on mice [27] and fish [28] subjected to continuous stress showed an increase in “Candidatus *Arthromitus*” in their intestinal microbiota. Our findings of “Candidatus *Arthromitus*” as a significant genus in songbirds in captive habitats, known for being stressful environments, contribute to the understanding of “Candidatus *Arthromitus*” as a genus marker for the compromised health status of animals under continuous stress [27,28].

In captive songbirds, we identified other genera, such as *Aeromonas*, *Empedobacter*, and *Acidibacter*, which have been described as responsible for pathologies. Importantly, to the best of our knowledge, none of these genera were previously associated with the conditions of songbirds. Members of the genus *Aeromonas* are highly associated with infectious conditions in fish and immunocompromised animal species [29]. On the other hand, *Empedobacter* species are related to human conditions such as periodontitis and meningitis [30], as well as infections and death in farmed fish [31]. Lastly, bacteria from the genus *Acidibacter*, identified as marker genera for captive songbirds, in addition to their pathogenic profile, may carry a high potential for antimicrobial multi-resistance [32,33].

The fecal bacterial community of captive songbirds includes certain genera recognized as environmental and beneficial bacteria, specifically *Acinetobacter*, *Flavobacterium*, and *Sphingobacterium*. The genus *Acinetobacter* is predominantly found in the microbiota of insects, such as cicadas, and is identified in bacteriome analysis of flowers, with its presence potentiated when added to fertilizers [34]. Interestingly, *Acinetobacter* abundance increases in insects that feed on fertilized plants [34]. *Acinetobacter* can also be found in natural water sources, contributing to biofilm formation [35,36]. Additionally, *Acinetobacter* has been isolated from soil [37], iron, and other metal mines [38].

On the other hand, the genus *Flavobacterium* primarily consists of environmental bacteria present in soil and water in forest environments or native areas [39,40]. Similarly, *Sphingobacterium* is a genus represented by many environmental species, having been isolated from soil and natural water sources [41,42].

The bacterial community data from fecal samples of wild songbirds revealed the presence of both potentially pathogenic and environmental bacterial genera in animals from this habitat. The genera identified as exclusive to wild songbirds were *Catelliococcus*, *Actinobacillus*, *Brevibacterium*, *Clostridium sensu stricto* 1, *Serratia*, and *Mycoplasma*. However, it is noteworthy that, unlike captive animals, our analyses did not identify any bacterial taxa as markers for wild animals. *Catelliococcus* was found to be the most abundant genus in wild songbirds, and it has been previously isolated from the intestinal microbiota of Thick-Billed Murre (*Uria lomvia*) [43] and passerines such as *L. megarhynchos* and *L. luscinia* [25], as well as from beach sand and seawater [44]. The presence of *Catelliococcus* in the feces of mammals and birds makes it a marker of fecal contamination in beach and lake waters [45,46]. The *Actinobacillus* genus has been associated with infections such as periodontitis, endocarditis, and meningitis in mammals [47,48]. It has also been isolated from *Anseriformes*, promoting respiratory diseases [49]. On the other hand, *Brevibacterium*, identified in the analyzed wild songbirds, is a genus primarily composed of environmental bacteria found in seas and rivers, known for their ability to secrete pigments and other substrates [50,51,52]. Interestingly, this genus has been identified in the microbiota of soft ticks of seabirds [53]. However, *Brevibacterium avium* is a potential cause of bumblefoot in poultry [54,55].

With beneficial effects, *Clostridium sensu stricto* 1 was identified in wild *S. similis*. These bacteria, involved in lipid metabolism, are used as probiotics for broiler chickens [56] and in fortifying solutions for premature babies [57]. Additionally, *Clostridium sensu stricto* 1 was previously described in the microbiome of passerines *L. megarhynchos* and *L. luscinia* [25].

Bacteria of the *Serratia* genus can naturally be found in the intestinal microbiota of some animals and in the environment. However, in cases of dysbiosis and immunological imbalances, *Serratia* can become highly pathogenic [58]. Some species of this genus also have zoonotic potential, such as *Serratia fonticola*, which is naturally present in the intestines of wild birds, and their feces serve as a source of contamination for humans [59].

Considering the members of the genus *Mycoplasma*, they have been isolated from wild songbirds in outbreaks of mycoplasmal conjunctivitis but, up until now, have not been observed in the microbiota of healthy songbirds [60]. On the other hand, *Mycoplasmas* were identified in the intestinal microbiota of the Passeriformes *L. megarhynchos* and *L. luscinia* [25].

Our results highlight the presence of fecal biomarker genera on the feces of animals subjected to stressful and unwanted living conditions in captivity. The identified fecal biomarkers could serve as measures of the quality of life for *S. similis* and as diagnostic markers for mucosal diseases. Furthermore, these genera markers can be used as potential indicators of environmental behavior.

In addition to “Candidatus *Arthromintus*” and *Acidibacter*, the other genera observed as fecal markers of *S. similis* in captivity were *Kocuria*, previously isolated from the preen glands and uropygial glands of owls [44], and *Paracoccus*. *Paracoccus* is already known as both a natural probiotic for some birds, influencing color and nest care behavior, and a genus involved in the metabolism of ammoniacal nitrogen and organic pollutants in poultry processing industrial effluent [61].

## 5. Conclusions

In conclusion, this study presented the first descriptive study of the fecal bacterial community composition of *S. similis* living in two different habitats (captivity and the wild). Our results suggest that the bacterial genera identified in the feces of animals from each habitat have specific evolutionary particularities and genetic characteristics. The lifestyle of *S. similis* birds significantly impacts the composition of the fecal microbiota, with probable impacts on the health and well-being of these birds. The bacteria biomarker identified in these animals can be used to establish the well-being of songbirds in captivity.

## Figures and Tables

**Figure 1 animals-14-00937-f001:**
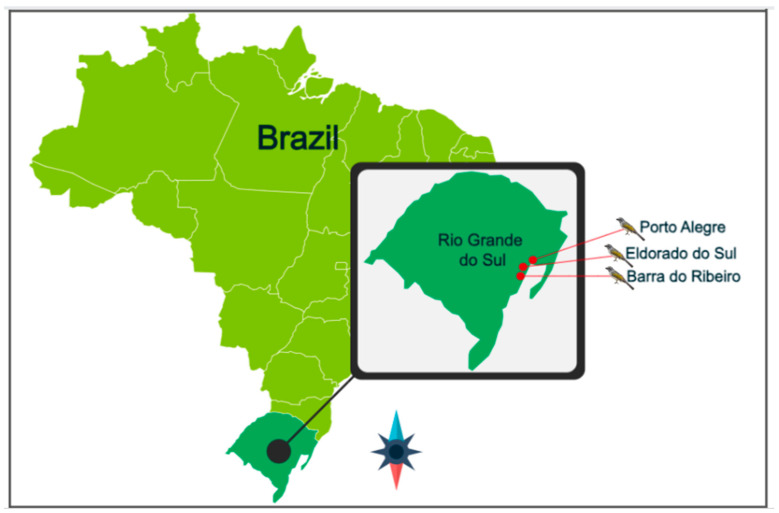
Illustrative demarcation of the collection area. Brazil—Rio Grande do Sul State. Red points: municipalities where fecal samples were collected from *Saltator similis*. Barra do Ribeiro and Eldorado do Sul: wild *Saltator similis*; Porto Alegre: captive *Saltator similis*.

**Figure 2 animals-14-00937-f002:**
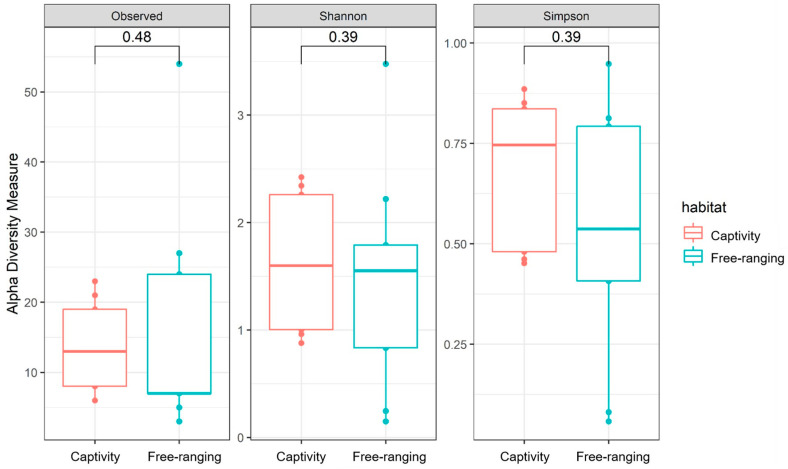
Alpha diversity analyses of fecal bacterial communities between captive and wild *Saltator similis*. Alpha analyses of Observed (*p* = 0.48), Shannon (*p* = 0.39), and Simpson (*p* = 0.39) metrics.

**Figure 3 animals-14-00937-f003:**
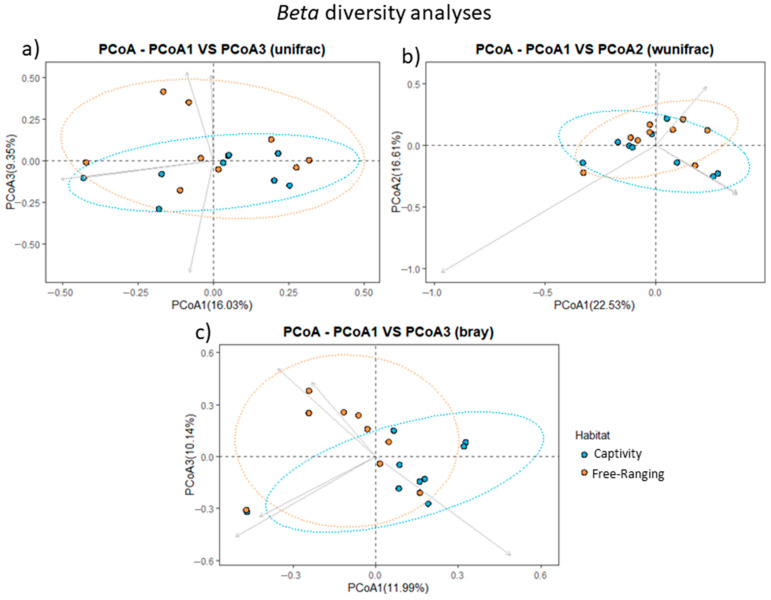
*Beta* diversity analyses of fecal bacterial communities between captive and wild *Saltator similis*. (**a**) Unweighted Unifrac. (**b**) Weighted Unifrac. (**c**) Bray–Curtis. Principal coordinate analysis plot (PCoA) of beta diversity representative of differences between groups utilizing UniFrac (PERMANOVA). The points represent each library. The blue points represent the songbirds from captivity, and the orange points represent the wild songbirds.

**Figure 4 animals-14-00937-f004:**
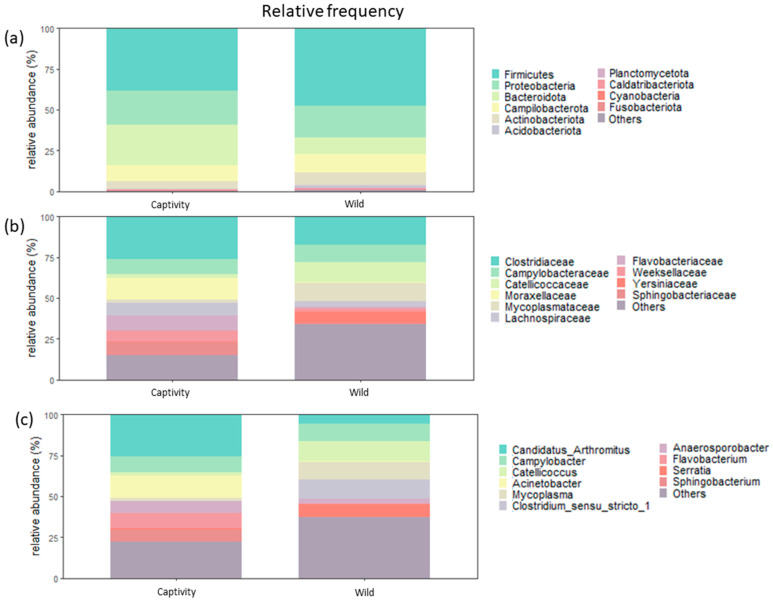
Relative abundance of overall phyla, family, and genera identified in the feces of the studied *Saltator similis*. Graphical representation of the relative number of reads found for (**a**) phylum, (**b**) family and (**c**) genera among songbirds in the wild and songbirds in captivity.

**Figure 5 animals-14-00937-f005:**
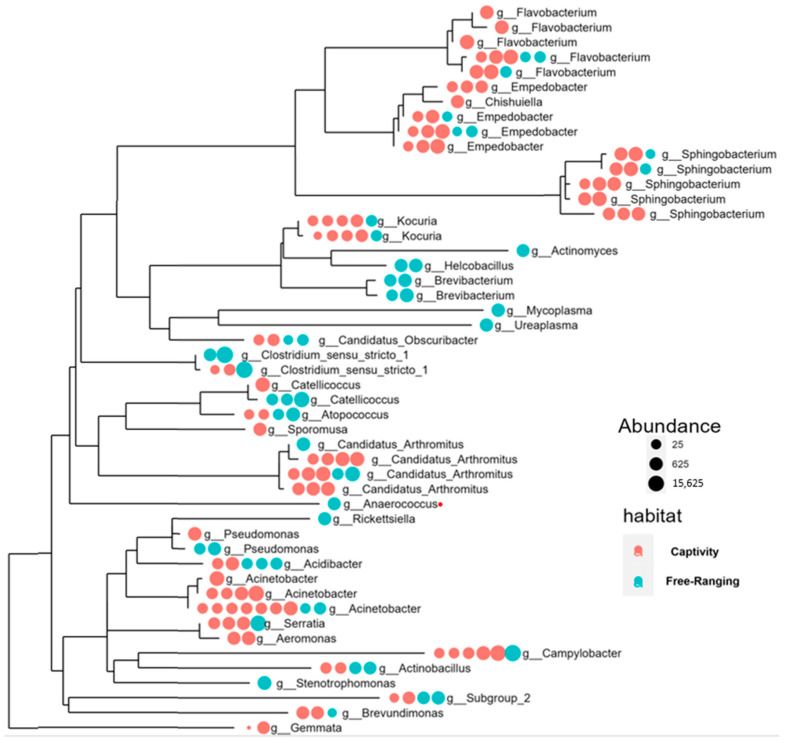
Evolutionary distribution of relative abundance of the genus identified in the feces of captive and wild *Saltator similis*. Pink points: captive songbirds. Turquoise points: wild songbirds. The size of each point represents the abundance of genus reads. Each point represents one library/sample. Results are from RStudio using packages TREEIO and Phyloseq.

**Table 1 animals-14-00937-t001:** *Saltator similis* sampled in this study and the library profile obtained in the NGS sequencing.

Animal Code	Origin (City)	Habitat	General Health Status	Life Stage	Raw Reads	Filtered Reads	Non-Chimeric Reads (%)
1	Barra do Ribeiro	Wild	Healthy	Adult	275,096	260,926	73.75
2	Barra do Ribeiro	Wild	Healthy	Adult	202,753	193,485	79.92
3	Barra do Ribeiro	Wild	Healthy	Adult	195,656	187,601	92.33
4	Barra do Ribeiro	Wild	Healthy	Adult	248,750	236,447	91.41
5	Barra do Ribeiro	Wild	Healthy	Adult	46,064	43,080	90.69
6	Eldorado do Sul	Wild	Healthy	Adult	184,185	175,746	50.38
7	Eldorado do Sul	Wild	Healthy	Adult	3713	2995	59.14
8	Eldorado do Sul	Wild	Healthy	Adult	253,852	243,964	80.79
9	Eldorado do Sul	Wild	Healthy	Adult	237,229	230,719	84.64
10	Porto Alegre	Captivity	Healthy	Adult	224,010	217,867	85.80
11	Porto Alegre	Captivity	Healthy	Adult	198,628	193,330	94.00
12	Porto Alegre	Captivity	Healthy	Adult	189,314	181,439	85.92
13	Porto Alegre	Captivity	Healthy	Adult	152,275	146,937	76.68
14	Porto Alegre	Captivity	Healthy	Adult	207,188	193,388	67.68
15	Porto Alegre	Captivity	Healthy	Adult	260,863	246,147	80.81
16	Porto Alegre	Captivity	Healthy	Adult	162,525	157,993	70.63
17	Porto Alegre	Captivity	Healthy	Adult	210,880	201,375	75.81
18	Porto Alegre	Captivity	Healthy	Adult	224,236	216,852	74.28

**Table 2 animals-14-00937-t002:** Genera biomarkers of the captivity habitat detected on bacterial communities from the feces of the analyzed *Saltator similis*.

Markers	Genus	Habitat	Effect Linear Discriminant Analysis	*p* Value	*p* Adjusted
Marker 1	“Candidatus *Arthromitus*”	Captivity	5,337,960	0.046972020	0.046972020
Marker 2	*Acinetobacter*	Captivity	5,236,298	0.001221026	0.001221026
Marker 3	*Kocuria*	Captivity	4,664,164	0.023832113	0.023832113
Marker 4	*Paracoccus*	Captivity	3,428,577	0.011922794	0.011922794

## Data Availability

The data are contained within the article.

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
