# Peer review of "Comparative Analysis of How the Fecal Microbiota of Green-Winged Saltator (Saltator similis) Diverge among Animals Living in Captivity and in Wild Habitats"

_animals, 2024, doi:10.3390/ani14060937_

Round 1

Reviewer 1 Report

Comments and Suggestions for Authors

Authors have characterized the gut microbiome of a songbird (S> similis) native to Brazil in captivity and free-range conditions. Overall, the authors have done a fine job in establishing the uniqueness of their research and discussing the implications of gut microbiome on health of these birds. I have following suggestions:

1. Were samples from these 2 groups taken in same weather conditions/season? Gut microbiome of wild animals have shown to be associated with seasonality.

2. How were the reads normalized/rarefied? Mentioned in results but add in methods too.

3. Similarly, which distance matrix were used? mentioned in results but add in methods too.

4. I am not sure how Shapiro-Wilk normality test was used to compare the differences in number of reads. I believe it is a test to see if the distribution deviated away from normal.

5. Please clean the taxa name in figure 4. 

Table 2: How was the Linear Discriminant analysis performed?

Author Response

Reviewer 1:

Comments and Suggestions for Authors

Authors have characterized the gut microbiome of a songbird (S> similis) native to Brazil in captivity and free-range conditions. Overall, the authors have done a fine job in establishing the uniqueness of their research and discussing the implications of gut microbiome on health of these birds. I have following suggestions:

Response: Thank you by your time and contributions with the manuscript.

  1. Were samples from these 2 groups taken in same weather conditions/season? Gut microbiome of wild animals have shown to be associated with seasonality.

Response: We appreciated the reviewer comment. Samples collection were performed in the same season. We have add this information in methods section. L. 84-86.

  1. How were the reads normalized/rarefied? Mentioned in results but add in methods too.

Response: Thank you by the observation. Information included in the methods section. L. 133-150.

  1. Similarly, which distance matrix were used? mentioned in results but add in methods too.

Response: Thank you by the observation. After numerous filtering and decontamination proccess, we worked with a remain percentage of at least 50% of non-chimeric reads, having a minimum size of 2,995 and a maximum of 260,926 bp. Information included in the methods section. L. 133-150.

  1. I am not sure how Shapiro-Wilk normality test was used to compare the differences in number of reads. I believe it is a test to see if the distribution deviated away from normal.

Response: We understand the concern. The Shapiro-Wilk is one of the distribution tests performed and is considered a test of normality. Shapiro-Wilk’s method is widely recommended for normality testing and it provides one of the most reliable. It is based on the correlation between the data and the corresponding normal scores.

  1. Please clean the taxa name in figure 4. 

Response: Thank you by the observation. Actually, the figure quality is low because it was copy inside word. In the separate Figure file a high quality is detached.

Table 2: How was the Linear Discriminant analysis performed?

Response: This Table was generated in RStudio using the Vegan, Adonis and Adonis2 packages for statistical and variance analyses, followed by the BiomarkeR package for genetic marker analyses.

Reviewer 2 Report

Comments and Suggestions for Authors

Specific results should be presented in the summary. The goal should be formulated based on the general biological significance of the species, with an understanding of what practical significance research can have.

1. What is the practical significance of the research?

2. How can the results obtained be useful in nature, influence processes, etc.?

3. Why was the SILVA database used when analyzing the sequencing results, where the data is irrelevant?

4. Why are NCBI or GTDB databases not used?

Author Response

Reviewer 2:

Comments and Suggestions for Authors

Specific results should be presented in the summary. The goal should be formulated based on the general biological significance of the species, with an understanding of what practical significance research can have.

Response: Thank you by your time and contributions with the manuscript.

  1. What is the practical significance of the research?

Response: Fecal microbiota of S. similis is influenced by the habitat. The animals living in captivity showed a dysbiosis in the microbiota, with some bacteria genera being indicated as biological markers to environmental behavior. Thus, the present research provides a new concept of life quality measure of songbirds.

We agree with the Reviewer that the mainly significance was not well-explored. Modifications were performed in L. 39-43 and L. 373-376.

  1. How can the results obtained be useful in nature, influence processes, etc.?

Response: We think that, as described in the previously question: the present research provides a new concept of life quality measure of songbirds.  Modifications were performed in L. 39-43 and L. 373-376.

  1. Why was the SILVA database used when analyzing the sequencing results, where the data is irrelevant?

Response: Current, taxonomic assignment of ASV, from 16S-amplicon sequencing, are commonly based on SILVA database.

  1. Why are NCBI or GTDB databases not used?

Response: NCBI directly is not a good approach, because the download database. We choose SILVA, because it is a database with high actualization and users around the world. 

Reviewer 3 Report

Comments and Suggestions for Authors

Dear authors,

in your manuscript you presented the investigation related to the fecal microbiota of wild and captive green-winged saltator.

Positive sides of your investigation are the use of modern molecular methods to describe the bacterial microbiome and a detailed analysis of the results, as well as comparison of wild and birds kept in captivity. Negative aspect is the low number of samples used in this investigation and data missing in describing of sampled animals.

Specific comments:

Title - please also mention English name of the bird 

L25-26 please rephrase this sentence ("Microbiota changes")

L 28 and in the whole text - consider to use word "wild" instead of "free-ranging"

Keywords - consider also microbiota, microbiome

Introduction - too short; general data about bird species using in this investigation could be added; some literature data about other findings on microbiota of similar birds species, et. 

L 47-49 please rephrase this sentence (adaptation, adaptive tool, quickly adapt)

Material and Methods - animal selection - please add mora data about samples birds (if known - sex and age, general health status)

You stated that there is notably high population of this bird species in the captivity (L61), so you could collect more samples from captive birds?

L 73 regarding captive birds - how long were those birds kept in this Wild Animals Triage Center; where they there due to some rehabilitation or illness?

L 90 please check this part of the sentence

Titles of sub-chapters - no need to write S. similis in every title

L 93 - amount of fecal sample of wild birds was also 5g? 

L106 reference not later found in the list of references

L146 please change "package Phylosec package"

L 149,152,15,170 please check writing of bird Latin name

L 166-168 please rephrase this sentence, specially last part

L 174-175 please rephrase this sentence (free-ranging bacterial community)

L 182, 198,199,215,217,223-227,236-237,247-248 not in italic

L 209-210 please rephrase this sentence (in animal abundances?)

L 231 in this title you mention "gut bacterial community" - in text up to now you used "feces or fecal bacterial community"?

L 2582 title should be changed

L341-342 please rephrase this sentence

L 349-350 please rephrase "protection of eggs"

According to the Instructions for authors of journal Animals, references in the text should be written as described here:

In the text, reference numbers should be placed in square brackets [ ], and placed before the punctuation; for example [1], [1–3] or [1,3]. 

Comments on the Quality of English Language

minor editing needed, a lot of bacterial genus or species are not written in Italic, etc.

Author Response

Reviewer 3:

Comments and Suggestions for Authors

Dear authors,

in your manuscript you presented the investigation related to the fecal microbiota of wild and captive green-winged saltator.

Positive sides of your investigation are the use of modern molecular methods to describe the bacterial microbiome and a detailed analysis of the results, as well as comparison of wild and birds kept in captivity. Negative aspect is the low number of samples used in this investigation and data missing in describing of sampled animals.

            Response: Thank you by your time and contributions with the manuscript.

We agree that the sample number is low. However, research with wild animals is very difficult to be performed. The access to the animals are very limited and expensive. Since with a low number, this is the first study with fecal bacterial community composition of S. similis. So, we believe that this data would be very useful to academic and non-academic readers.

Specific comments:

Title - please also mention English name of the bird 

Response: We appreciated the suggestion. Modifications were performed.

L25-26 please rephrase this sentence ("Microbiota changes")

Response: We appreciated the observation. Modifications were performed.

L 28 and in the whole text - consider to use word "wild" instead of "free-ranging"

Response: We appreciated the suggestion. The word was changed in the entire manuscript.

Keywords - consider also microbiota, microbiome

Response: We thanks by the suggestion.

Introduction - too short; general data about bird species using in this investigation could be added; some literature data about other findings on microbiota of similar birds species, et. 

            Response: We agree with the reviewer. Modifications L. 55-66.

L 47-49 please rephrase this sentence (adaptation, adaptive tool, quickly adapt)

Response: Modifications L. 52-53

Material and Methods - animal selection - please add mora data about samples birds (if known - sex and age, general health status)

Response: We agree with the reviewer. Modifications L. 89-93.   

You stated that there is notably high population of this bird species in the captivity (L61), so you could collect more samples from captive birds?

Response: Despite there being a large number of captive specimens, the majority of these animals present sequelae from animal trafficking, nutritional dysfunctions or diarrhea caused by protozoa or bacteria and undergo treatment with antibiotics. Avoiding any type of interference, we sought to select healthy, adult animals without gastrointestinal disorders, which significantly restricted our “N”. Furthermore, we sought to maintain the captive population proportional to the wild population in order to obtain reliable statistical results and truly reliable biomarkers.

L 73 regarding captive birds - how long were those birds kept in this Wild Animals Triage Center; where they there due to some rehabilitation or illness?

Response: The Wild Animals Triage Center is a place that held wild animals of all types, from rehabilitation animals to victims from animal trafficking. The animals selected for this study were heath animals, that were seized at IBAMA and were there awaiting legal action. After the release of the law, if suitable, they will be prepared for release.

L 90 please check this part of the sentence

Response:  Modifications were performed.

Titles of sub-chapters - no need to write S. similis in every title

Response: Thank you. Modifications were performed.

L 93 - amount of fecal sample of wild birds was also 5g? 

            Response: Yes. Description in L. 111.

L106 reference not later found in the list of references

            Response: Thank you by the observation.

L146 please change "package Phylosec package"

            Response: Modifications L. 169

L 149,152,15,170 please check writing of bird Latin name

Response: Modifications performed.

L 166-168 please rephrase this sentence, specially last part

            Response: Modifications L. 170-172  

L 174-175 please rephrase this sentence (free-ranging bacterial community)

            Response: Modifications L. 194-200

L 182, 198,199,215,217,223-227,236-237,247-248 not in italic

            Response: Thank you.

L 209-210 please rephrase this sentence (in animal abundances?)

            Response: Modifications L. 238-239

L 231 in this title you mention "gut bacterial community" - in text up to now you used "feces or fecal bacterial community"?

            Response: Thank you by the observation. Is “ fecal”.

L 2582 title should be changed

            Response: Modification performed

L341-342 please rephrase this sentence

            Response: Modifications L. 362

L 349-350 please rephrase "protection of eggs"

            Response: Modifications L. 370-371

According to the Instructions for authors of journal Animals, references in the text should be written as described here:

In the text, reference numbers should be placed in square brackets [ ], and placed before the punctuation; for example [1], [1–3] or [1,3]. 

 Response: Thank you by the observation. References were adequate.

Round 2

Reviewer 3 Report

Comments and Suggestions for Authors

Dear authors,

thank you for taking into consideration all the comments given by the reviewer.

Best regards

Comments on the Quality of English Language

minor editing needed